# New Perspectives for Whole Genome Amplification in Forensic STR Analysis

**DOI:** 10.3390/ijms23137090

**Published:** 2022-06-25

**Authors:** Richard Jäger

**Affiliations:** 1Department of Natural Sciences, Bonn-Rhein-Sieg University of Applied Sciences, von-Liebig Str. 20, 53359 Rheinbach, Germany; richard.jaeger@h-brs.de; 2Institute for Functional Gene Analytics, Bonn-Rhein-Sieg University of Applied Sciences, Grantham Allee 20, 53757 Sankt Augustin, Germany; 3Institute of Safety and Security Research, Bonn-Rhein-Sieg University of Applied Sciences, Grantham Allee 20, 53757 Sankt Augustin, Germany

**Keywords:** short tandem repeat (STR), forensic genetics, DNA typing, whole genome amplification (WGA)

## Abstract

Modern PCR-based analytical techniques have reached sensitivity levels that allow for obtaining complete forensic DNA profiles from even tiny traces containing genomic DNA amounts as small as 125 pg. Yet these techniques have reached their limits when it comes to the analysis of traces such as fingerprints or single cells. One suggestion to overcome these limits has been the usage of whole genome amplification (WGA) methods. These methods aim at increasing the copy number of genomic DNA and by this means generate more template DNA for subsequent analyses. Their application in forensic contexts has so far remained mostly an academic exercise, and results have not shown significant improvements and even have raised additional analytical problems. Until very recently, based on these disappointments, the forensic application of WGA seems to have largely been abandoned. In the meantime, however, novel improved methods are pointing towards a perspective for WGA in specific forensic applications. This review article tries to summarize current knowledge about WGA in forensics and suggests the forensic analysis of single-donor bioparticles and of single cells as promising applications.

## 1. Introduction

This review focuses on the suitability of whole genome amplification (WGA) for forensic DNA profiling that uses current standard technologies. To be able to appreciate both the possible applications and the limitations of WGA in forensic DNA analysis, it will first be necessary to explain the basics of forensic DNA profiling from which the obvious fields of application of WGA will be motivated. Then, the technical principles of WGA will be described in conjunction with a critical discussion of published work applying WGA in forensics. By this means, the common deficiencies of WGA in a forensic context will be established, which finally will lead to the identification of fields of application where improved WGA methods may be promising.

## 2. Forensic DNA Profiles

Forensic DNA profiles are based on the analysis of a standardized set of short tandem repeat (STR) loci that are highly polymorphic in the human population. In the European Union, 15 different loci are currently analyzed; in the North American CODIS system, this standard set is complemented by five additional loci [1,2]. STR loci are characterized by multiple repeat units of few nucleotides that are arranged in a tandem fashion one after the other. With few exceptions, the STR loci analyzed in human forensics have repeat units consisting of four nucleotides [3]. The alleles of the STR loci differ in the number of repeat units, which may amount to several dozens, depending on the STR locus. The allele designations simply represent the numbers of repeat units as related to standard alleles [4]. The number of repeat units corresponds to a length in base pairs and can thus be determined by electrophoresis following PCR using primers that bind in the conserved regions flanking the tandem repeats. Depending on the locus, nine to over forty different alleles can be identified, and the combination of resulting genotypes makes such an STR profile statistically unique within the human population [3].

Technically, the set of forensic STR loci is analyzed using multiplex PCR, and the amplified fragments are sized using capillary electrophoresis (CE) [5,6]. One of the two primers amplifying each locus is labeled with a fluorophore, allowing for detection. Four (or five) different fluorophores are assigned to the various loci in such a way that fragments of each STR locus can be unequivocally identified based on size and color on the electropherogram. A heterozygous genotype of a particular locus thus will display two peaks of similar height on the electropherogram, whereas a homozygous genotype will display one peak (Figure 1).

In modern forensic STR typing, commercial reagent kits are used that have been validated on the commonly used CE devices. Sizing standards and allele standards included in these kits allow for semi-automatic evaluation of electropherograms by software that assigns allele numbers to peaks for each locus. Of particular importance for this process of “allele calling” are threshold settings which preclude allele assignment for peaks that result from analytical noise or from typical technical artifacts [7].

One important type of technical artifacts that typically occur in STR analysis are so-called stutter peaks, which result from the propensity of the repeat units to slip by one or more units during the elongation step of PCR amplification [8,9]. As a consequence, stutter peaks are seen as small peaks preceding the main peaks and are typically one complete repeat unit shorter than the true alleles (Figure 1). Stutter peaks with sizes that are one unit longer or two or more repeat units different from the main peak sometimes occur as well. The incidence of replication slippage in a particular PCR assay is characteristic for each locus, and thus either a general stutter threshold or locus-specific stutter thresholds are applied [10]. Preclusion of stutter peaks is important because they have the same lengths as expected for true alleles and would lead to wrong interpretations of electropherograms.

## 3. Analytical Challenges and the Prospects of WGA

### 3.1. Stochastic Effects and Low Template DNA

Modern commercial STR kits are highly sensitive and can establish full profiles from as little as 125 pg of genomic DNA [11,12,13]. (A human cell contains 6.6 pg of nuclear DNA). At lower concentrations, single alleles or loci may escape detection. Even running more PCR cycles will not overcome this limit, which shows that it is not just analytical in nature. Rather, the limit reflects the occurrence of stochastic sampling effects [14]. These may have two explanations: First, in a trace with a DNA amount corresponding to only few genomic copies, some DNA loci may be present in unequal abundancies. Second, if in a DNA sample only few genomic copies are present, any fraction to be analyzed may no longer represent the complete genome. As a consequence, alleles will be underrepresented or absent, resulting in typical stochastic effects on electropherograms (Figure 1), such as pronounced peak height imbalances between loci or between the two alleles of one locus (allelic imbalances, AI), or allele peaks being completely missing (allele drop-out, ADO) [15]. Low DNA amounts analytically entailing stochastic effects are referred to as low template DNA (LT DNA) (also termed low copy number DNA, LCN DNA) [16] and warrant more sensitive analytical procedures, which in turn may evoke additional artifacts, such as allele drop-ins (ADI). On electropherograms, ADIs present as peaks that resemble normal allele peaks and may result from the amplification of contaminants (present in the sample, in the equipment or in reagents). Furthermore, they may be due to replication slippage events occurring during early PCR cycles when still only a few template molecules are present, such that resulting stutter fragments may become prominent [17].

To comply with stochastic effects, LCN DNA methods involving lower reaction volumes and more PCR cycles are applied in two or three replicates in order to identify those peaks as reliable that are reproduced by at least two assays [16]. This common way of analysis has been criticized because dividing a sample with already limiting amounts of DNA may even exacerbate the stochastic sampling effects; thus, information could be better obtained by analyzing the complete sample in one assay [18]. A disadvantage of the latter strategy is, however, that stochastic effects cannot be identified as such, and replicate analysis is not possible [19]. In this context, the application of WGA would offer the advantage of generating larger amounts of template that would allow replicate analysis without the risk of eliciting additional stochastic effects due to further dilution of the sample.

The differences between WGA and simply increasing the cycle number of an ordinary PCR are not immediately obvious, and if DNA loci are missing right from the start, WGA will not be able to overcome resulting stochastic effects. However, WGA may reduce the risk of generating unspecific amplification products. Similarly to nested PCR protocols [20], the first rounds of amplification are performed with different PCR primers than used in the actual STR analysis.

### 3.2. Degraded DNA

Most DNA-containing traces have been exposed to the environment; thus, DNA integrity may be compromised by environmental influences, such as humidity, heat, acidic or oxidizing conditions, UV exposure, or enzymatic degradation (reviewed in [21]). Typically, this results in damaged or fragmented DNA, which precludes PCR amplification of affected DNA loci. As a consequence, ADOs may occur and may encompass whole loci (locus drop-out, LDO). As the chance of experiencing damage is proportional to the length of a DNA molecule, DNA degradation typically affects the longer PCR amplicons first, resulting in more pronounced reductions in peak heights and increasing appearances of ADOs and LDOs on the right side of an electropherogram (corresponding to longer DNA framents). If DNA damage is too severe, all STR loci will be affected, and peaks will remain below the detection threshold.

As DNA degradation will result in a lower number of copies of STR loci that can be amplified, one strategy might consist in the preamplification of genomic DNA by WGA in order to increase the number of the few copies that are still intact. To these ends, the suitability of WGA methods for environmentally exposed DNA traces needs to be evaluated

## 4. WGA Methods Tested in Forensic STR Analysis

### 4.1. Overview

WGA methods can be classified as PCR-based or as based on multiple displacement amplification (MDA) [22]. In addition, there are methods that do not conform to this distinction, as they either combine both principles or are based on non-related principles [23,24,25].

PCR-based principles use either mixtures of primers with randomized sequences that can bind to many DNA loci, and by this means in theory will amplify all parts of the genome (Figure 2A) [26,27], or are based on targeted or random fragmentation of genomic DNA followed by ligation of adaptor oligonucleotides at the fragment ends that allow for PCR amplification with optimized primers that are complementary to adaptor sequences (Figure 2B) [28,29,30,31].

MDA is based on the high-fidelity DNA polymerase phi29 that exhibits high processivity (synthesizing continuous DNA of up to 20 kb) and has strand displacement activity [32]. MDA-based WGA begins with denaturation of the genomic DNA and elongation of the complementary strands after annealing of short primers with randomized sequences (typically hexameric). When the polymerase reaches a double-stranded region that has already been synthesized by another phi29 polymerase acting on the same strand, the preceding strand will be displaced, generating a novel single-stranded template for further random-primed elongation. By this means, the template will be multiplied in a quasi-exponential fashion by generating arborized DNA template arrays (Figure 2C).

### 4.2. Basic WGA Methods and Variations

#### 4.2.1. DOP-PCR

Degenerate oligonucleotide primed-PCR (DOP-PCR) was among the first WGA protocols developed [26]. It is a PCR-based method that uses primers with six random nucleotides embedded between defined short sequences on either end, which are first used for few PCR cycles at low stringency (to allow amplification of multiple regions of the genome), followed by a larger number of high-stringency amplification cycles for specific enrichment of the products (Figure 2A). For STR analysis, success rates between 50% and 75% have been reported for DNA amounts lower than 60 pg (12 STR loci tested) [33]. The protocol has been modified several times by altering the primer sequences, cycle numbers, and DNA polymerases in order to improve genome coverage and STR typing success rate, leading to improved methods called LL-DOP-PCR [34], dcDOP-PCR [35], mDOP-PCR [36], and iDOP-PCR [37].

#### 4.2.2. PEP PCR

Differently from DOP-PCR, primer-extension-preamplification PCR (PEP PCR) uses primers of 15 nucleotides that are completely degenerate, and amplification starts at low-stringency annealing temperatures which are continuously raised in the subsequent PCR cycles (Figure 2A) [27]. The method worked for single-cell analysis and was subsequently further optimized in terms of PCR cycle parameters and DNA polymerases to improve genome coverage and the success rate of STR analysis of clinical samples (I-PEP PCR) [38], and after further modification, of forensic samples (mIPEP PCR) [39].

#### 4.2.3. Adaptor Ligation-Mediated PCR

Adaptor ligation-mediated PCR methods (Figure 2B) differ in the way that the random DNA fragments are generated. The initial method used the restriction enzyme Mse1 to introduce cuts in the template DNA, followed by linker annealing and ligation to the generated fragments [30]. The commercial Omniplex and GenomePlex methods are based on chemical fragmentation of the template DNA [28,29], whereas another method called PSRG (adaptor–ligation PCR of randomly sheared genomic DNA) generates DNA fragments by hydrodynamical shearing of genomic DNA, followed by fill-in of resulting overhangs and adaptor ligation [31].

#### 4.2.4. MDA

Most MDA methods rely on phi29 polymerase in conjunction with random primers. Related methods use a primase instead of random primers that is coupled to a DNA polymerase with strand displacement activity [40,41]. Generally, MDA methods require long, uninterrupted template DNA sequences and tend to underrepresent the ends of template DNA fragments (Figure 2C). As a means to make MDA suitable for the fragmented DNA often seen in forensic DNA samples, protocols have been proposed that circularize the template DNA fragments, allowing for rolling circle amplification using the MDA principle (Figure 2D). In the RCA-RCA WGA protocol (developed for DNA from formalin-fixed tissue) DNA fragments generated by restriction enzyme are circularized by self-ligation, followed by exonuclease digestion of the remaining linear fragments [42]. A related protocol termed blunt-end ligation-mediated WGA (BL-WGA) has been established for plasma-circulating DNA fragments, the ends of which are first blunted by T4 polymerase and then ligated using T4 ligase, generating circular substrates and concatemers, which are then subjected to phi29-mediated rolling circle amplification and MDA [43].

### 4.3. Limitations of WGA in Forensic STR Analysis

#### 4.3.1. A Priori Limitations of the WGA Methods

All WGA methods tend to display some bias in terms of amplification of genomic DNA loci (reviewed in [44]); thus, the uniformity and completeness of the genome coverage of WGA products are critical parameters when it comes to downstream analysis of multiple DNA loci (Figure 3). In forensic STR analysis of LT DNA, WGA-inherent bias thus bears the risk of generating additional ADOs and AIs on top of the already present stochastic artifacts. Forensic STR analysis imposes two further challenges: impaired integrity of template DNA and the propensity of STR loci for replication slippage during PCR amplification.

Generally, PCR-based methods can better deal with low quality DNA (damaged or fragmented), because unlike MDA, PCR does not rely on long, undisrupted templates; however, the rolling circle MDA variants are suitable for fragmented DNA templates too. Like normal PCR amplification, the PCR-based protocols are prone to stutter artifacts when analyzing STR loci [45], whereas replication slippage is less likely to occur in MDA-based methods [46].

The adaptor ligation-mediated PCR methods exhibit a fundamental problem if applied to only few copies of template DNA. Unlike the random primers used in DOP-PCR, PEP PCR, or MDA, which allow for initiating DNA synthesis multiple times from various locations without harming the template, the fragmentation of template DNA is irreversible, and thus any STR amplicon disintegrated during fragmentation (or located within a fragment too long for successful PCR amplification) will inevitably be underrepresented later on. This effect will be particularly apparent in the analysis of single cells where each STR allele is present only once.

#### 4.3.2. Experimentally Established Performances and Limitations of the WGA Methods

Already early in their development, the potential of WGA methods for forensic DNA analysis has been recognized [27,33], and methods have been evaluated in terms of STR analysis. Initial studies, however, did not analyze forensic standard loci, and they did not use forensically relevant DNA samples, such as extracts from typical trace types (saliva, semen, blood) or degraded DNA (artificially degraded or environmentally exposed).

Several later studies have then tested the various available WGA methods for their potential to improve the STR typing success of problematic DNA samples using contemporary commercial kits for forensic STR analysis. These studies are generally difficult to compare, because they analyze DNA from different sources and of different amounts, and often evaluate STR typing success and sensitivities in different ways. Moreover, the contemporary STR typing kits used differ in their sensitivities, which sometimes do not reach the limits of the STR kits nowadays in use. In this section, the most significant findings will be summarized, paying attention in particular to sensitivity of the methods, technical artifacts, and success in typing degraded DNA samples.

Two studies reported sensitivities down to to 10 pg input DNA for iPEP, GenomiPhi MDA, and the commercial adaptor ligation method GenomePlex [28,47]; however, they noticed the occurrence of AIs and ADOs at these extremely low DNA amounts. One study did not find an improvement for DOP-PCR, MDA, or I-PEP PCR over non-WGA for treated DNA (with LL-DOP-PCR completely failing), and as a consequence developed the mIPEP PCR, which was successfully applied to 5 pg DNA from buccal swabs, to semen stains, to vaginal swabs, and even to fingerprints [39]. However, the occurrence of ADIs and AIs was reported for mIPEP PCR using low DNA amounts, and the method was of little benefit when analyzing environmentally exposed bloodstains, suggesting deficiencies when applied to real-world forensic trace material [39]. A later study could obtain partial STR profiles from environmentally exposed blood stains using mIPEP PCR; however, it reported extra alleles (ADIs) with low amounts of template DNA [48]. In one study, DOP-PCR failed with low DNA amounts, and an improved DOP-PCR method (called iDOP-PCR), while achieving sensitivity down to 15 pg, showed high proportions of ADOs (46%) and ADIs (4%) [37]. Likewise, adaptor ligation PCR protocols, while generally improving the sensitivity of STR analysis, resulted in significant AIs, ADOs, and ADIs [28,29,31,37].

On the other hand, adaptor ligation PCR seems best suited for analyzing degraded DNA samples, as shown in two studies comparing STR typing success after the application of PEP PCR, DOP-PCR, adaptor ligation PCR, two MDA protocols, and the rolling circle MDA methods, to DNA extracted from heat-treated human muscle samples [49,50]. With the exception of PEP, all methods failed when analyzing DNA amounts of less than 1 ng, and only PEP and the adapator ligation method (GenomePlex) improved the typing success for degraded DNA; GenomePlex, however, generated many ADIs and high stutters [49,50]. Likewise, Uchigasaki et al. (2018) reported improved allele recovery after GenomePlex WGA applied to UV-irradiated human bloodstains; however, the observed peaks were different from those of the control samples [51]. Remarkably, in the studies of Maciejewska et al., the rolling circle MDA methods (initially developed for fragmented DNA) proved less successful compared with other WGA methods [49,50], confirming the findings of two earlier studies [39,52]. Ambers et al. (2016) modified the DOP-PCR protocol to improve the analysis of ancient and degraded forensic DNA samples. While their mDOP-PCR protocol improved STR typing success, they noticed the occurrence of artifacts such as ADOs, ADIs, and increased stutter [36]. In a recent study, a workflow was suggested for STR analysis of UV-exposed DNA samples [53]. The workflow incorporated mIPEP PCR, which was shown to improve allele recovery for low amounts of damaged DNA; however, it increased the number of ADIs.

To summarize these studies, WGA methods when applied to forensic STR typing, while generally increasing the analytical sensitivity, impose several novel problems: Profiles often display pronounced imbalances and ADOs that affect STR loci and alleles in a non-predictable fashion and are related to WGA-inherent bias. In addition, particularly with the PCR-based methods, high rates of stutters and ADIs are seen. These phenomena render STR profiles from unknown donors hard to interpret and lower the statistical power of the evidence. For example, in the case of an apparently homozygous locus (showing just one peak on the electropherogram), it cannot be decided whether a second allele is actually missing. ADIs or high stutters cannot be distinguished from normal alleles, and may thus mislead the interpretation as well. Furthermore, pronounced intra- or inter-locus peak height imbalances hamper the interpretation of mixed profiles, because peak heights no longer reflect the true amount of template DNA. Thus, although additional genotype information from LT DNA or damaged DNA can be obtained by WGA, the generated artifacts may mislead the interpretation of STR profiles, which strongly argues against the use of WGA in forensic casework.

## 5. Perspectives for WGA in Forensic STR Analysis

In light of the inability of WGA to significantly improve the STR typing of the typical forensic DNA samples, Barber and Foran (in a study comparing MDA and I-PEP PCR) in 2006 concluded that, “WGA appears to be of limited forensic utility unless the samples are of a very high quality” [54]—which, however, would make the use of WGA unnecessary. Publications in the following years have not led to a substantial revision of that judgement. Remarkably, no study has addressed the sensitivities of WGA methods against PCR inhibitors—compounds, such as heme, humic acid, and denim dyes, which are often coextracted with the DNA from traces and will impair PCR amplification by various mechanisms [55]. The ability to deal with PCR inhibitors is an important aspect in the developmental validation of forensic PCR assays (see, for example, [11]); however, the WGA methods have never been systematically tested in that respect. Thus, it seems forensic researchers have lost their enthusiasm for applying WGA to casework. This is the more true nowadays, as modern multiplex STR kits have improved sensitivities down to 60 pg input DNA [56], and LCN DNA methods based on them have greatly improved the analysis of trace DNA [57], removing the need to take the risk of additional WGA-caused bias and artifacts.

In the meantime, however, novel WGA protocols have been developed, aiming in particular at sensitivities on the single-cell level, and at the same time reducing amplification bias; and several WGA kits have been commercialized that have been optimized for single-cell analysis, particularly for usage in clinical settings. These novel methods and kits have again sparked interest in the application of WGA in forensic DNA analysis, but so far only few of them have been tested in a forensic context.

### 5.1. WGA Methods with Reduced Bias

The ADOs, LDOs, and pronounced AIs reported after WGA-based preamplification were observed at DNA amounts well above the stochastic threshold, and thus cannot be fully explained by stochastic sampling effects. Rather, they point towards amplification bias, which is typical of WGA applied to low template DNA concentrations, as random events during the initial amplification become exacerbated due to the exponential nature of the amplification process [44].

Several WGA protocols have been established that aim at reducing bias by including non-exponential amplification steps. Among those low-bias methods are the multiple annealing and looping based amplification cycles (MALBAC) method, which uses the *Bst* polymerase during a first quasi-linear preamplification step, preceding the subsequent PCR amplification [25], and the commercial SurePlex/PicoPLEX kit that is based on a related principle [24]. The LIANTI (linear amplification via transposon insertion) method is based on transposon-mediated generation of random genomic fragments with terminally attached T7 promoter sites that can be linearly transcribed into RNA capable of self-priming for subsequent DNA synthesis [23]. Recently, the PTA (primary template-directed amplification) method has been published, in which the phi29-polymerase-mediated extension of randomly primed DNA products is limited to short lengths, thereby preventing exponential amplification [58].

Two studies have been published that tested MALBAC in conjunction with forensic STR typing kits, both analyzing DNA extracted from human peripheral blood [59,60]. Even with this presumably high-quality DNA, both studies disappointed in terms of forensic STR analysis. In their study in 2022, Liao et al. noticed improved allele recovery (as compared to non-WGA samples) after the application of MALBAC to DNA amounts less than 50 pg. However, a high number of ADOs occurred, and profiles displayed many imbalanced STR loci and ADIs [59]. Likewise, a second study [60] comparing MALBAC WGA with a commercial single-cell MDA kit (Repli-g) and non-WGA-treated DNA, reported a higher number of called alleles after MALBAC and MDA for DNA amounts of less than 50 pg (although only less than 50% of alleles were called with either method). However, the percentage of erroneously called STR loci was significantly higher in MALBAC-amplified profiles, but it was not further specified how much ADOs or ADIs may have accounted for the errors. The increased occurrence of ADOs and ADIs after MALBAC as compared to MDA may be due to the *Bst* polymerase that was reported to be less sensitive and more prone to stutters than phi29 polymerase when amplifying STR loci [61].

High proportions of ADOs and ADIs were also reported for the methodically related PicoPLEX kit when applied to single cells [62]. Thus, although the polymerase used in the pre-amplification step of the PicoPLEX kit has not been disclosed, the two related low-bias methods, MALBAC and PicoPLEX, are most likely not suited for forensic STR analysis. The other low-bias methods, LIANTI and PTA, however, may still hold promise and would be worth testing in a forensic context.

### 5.2. WGA in the STR Analysis of Single Cells

#### 5.2.1. Micromanipulation of Single Cells and of Bioparticles for Mixture Deconvolution

With the high sensitivity of modern STR typing kits, the occurrence of mixed DNA profiles derived from more than one individual has increased, because now minute DNA amounts can be detected that have been left by other individuals who may not even be related to the actual crimes [63]. STR profiles of such mixtures typically display more than two peaks per STR locus [64]. Even DNA transferred indirectly may become detectable and confound the DNA profile of a perpetrator [63,65].

There are various ways to deconvolve the peaks on electropherograms of mixed STR profiles (i.e., assign them to individual donors), and modern software-assisted methods have increased the statistical power of the confounded information [66]. To be able to deconvolve the electropherograms of mixed profiles, it is of importance that on electropherograms the peak heights reflect the amounts of DNA from the respective donor individuals. Despite sophisticated software tools being available, however, interpretations of mixed STR profiles often remain unsatisfactory, particularly if peaks of the different contributors have similar heights or if stochastic effects confound the information [66].

As a way of avoiding mixtures right from the start, methods have been suggested that physically deconvolve mixed trace material by isolating bioparticles (such as skin flakes, and aggregates of a few cells or single cells) that contain the genomic DNA of exactly one donor individual [67,68]. The price to be paid is an extremely low amount of DNA, which necessitates LCN DNA methods entailing stochastic effects, particularly when analyzing replicates.

The feasibility of micromanipulating and genotyping single cells from forensic trace material, such as chewing gums, cigarette butts, swabs, touched skin, and fabrics, has been demonstrated in several studies [69,70,71,72,73]. These studies recovered single cells or small bioparticles containing the DNA from individual donors, and by this means established single-donor STR profiles using LCN DNA methods. A high proportion of single-donor profiles, however, were incomplete and also contained ADIs, and thus replicates of several such profiles had to be combined to establish the full profiles.

In the studies by Li et al. [71,72], buccal cells were micromanipulated from trace material, and using low-volume PCR, consensus profiles could be obtained by combining the profiles from five or six single mucosal cells. The used microwell slides are, however, no longer commercially available, and there have been no follow-up reports using this technology in forensics. The study by Farash et al. (2018) [74] described the analysis of micromanipulated cells or cell aggregates from skin deposited on touched materials. In about one third of the samples analyzed, STR profiles attributable to donors were obtained. The study by Ostojic et al. (2021) [73] compared several micromanipulation methods and could show that ten micromanipulated cells were sufficient to compile forensically informative profiles. A study by Huffmann (2021) showed that the application of an improved LCN DNA method to 1–3 cell subsamples of two-person mixtures allowed for successful compilation of consensus profiles of the contributors, with significant ADO and ADI rates in the individual profiles, however [70]. Based on these experiments, a suitable strategy for analysis of complex mixtures using software-assisted mixture analysis has been published recently, showing that analyzing several subsamples consisting of one to two cells can increase the statistical power as compared to analyzing bulk mixtures [75]. However, despite using LCN DNA methods, locus-specific drop-out rates were on average 58% for single-cell and 38% for two-cell subsamples [75].

Thus, if in a forensic context, single-cell WGA methods were able to deliver on their promise, i.e., to enable the genotyping of single cells, their application might lead to a further improvement by increasing the template DNA of one- or two-cell subsamples to amounts that can more reliably be analyzed with modern forensic STR kits, even allowing for replicate analysis of the subsamples.

#### 5.2.2. Forensic STR Analysis of Single Cells

The principal suitability of single-cell WGA methods for forensic STR analysis of single cells was tested in several recent studies. Analyzing single cells has the advantage of stochastic sampling effects being less likely, because from whole cells, complete diploid genomes can be extracted and then be subjected to WGA. In modern single-cell WGA kits, this is accomplished by carrying out cell lysis, DNA extraction, and WGA in the same tube.

In a study from 2018, the low-bias single-cell method PicoPLEX was compared to a commercial single-cell DOP-PCR kit (DOPlify), a single-cell MDA kit (Repli-g), and an adaptor ligation method (Ampli 1) [76]. In that study—analyzing genomic DNA from micromanipulated single cells from a human B lymphoblastoid cell line—the PCR-based methods caused the highest numbers of ADOs and LDOs, and PicoPLEX showed many LDOs and ADOs when applied to single cells. Though in that study ADIs were not addressed, another study applying the PicoPLEX kit to DNA from single unfixed or formalin-fixed cells, reported a frequency of ADIs of 11.6% [62]. Thus, PicoPLEX remained unsatisfactory in terms of STR analysis of single cells, whereas the single-cell MDA method Repli-g was promising.

The Repli-g single-cell MDA kit was tested in two further studies for its suitability for forensic STR typing. The study by Maruyama et al. (2020) reported that at least 20 micromanipulated buccal cells were required for successful STR-typing, whereas from single cells, most alleles remained undetected [77]. Another study by Chen et al. (2020), however, demonstrated that Repli-g single-cell WGA indeed allowed for successful STR analysis of single, micromanipulated B-lymphoblastoid cells; most single cells yielded complete profiles [78]. Intra- and interlocus peak height imbalances, however, were pronounced but became less so when analyzing three or five cells. Likewise, stutters were increased as compared to STR profiles from control DNA. In the study by Maruyama et al. (2020) [77], cells were dried on the applicator tips prior to micromanipulation, which may have affected DNA extraction or the integrity of cell nuclei. As a further variable, the volume of buffer cotransferred by the micromanipulation capillary may have accounted for the differences in STR typing success between the two studies, as this may lead to dilution or pH change of the extraction buffer.

#### 5.2.3. WGA in the Analysis of Single Sperm Cells

A recently emerging, potential forensic application of WGA is in the STR analysis of micromanipulated single sperm cells. In rape cases, the DNA from vaginal swabs will in most cases be derived from both sperm cells from the perpetrator and vaginal cells from the victim, and differential extraction protocols aiming at separating the sperm fraction from the victim DNA fraction often remain unsatisfactory [79]. The analysis of single sperm cells micromanipulated from vaginal swabs can thus be considered a special application of physical mixture deconvolution that might even help in the analysis of traces with low sperm count and in the clarification of multiple perpetrator rape cases.

Studies addressing single sperm cell analysis using conventional forensic STR analysis, however, reported that at least 20 sperm cells are required to establish complete STR profiles [80,81,82]. Sperm cells are haploid, and based on statistical considerations, a minimum of nine single sperm cells is required to compile a diploid profile [83]. However, even haploid STR profiles of single sperm cells may already be attributable to individual donors and thus be helpful in the clarification of crime cases. One of the disadvantages of WGA, the occurrence of allelic imbalances, is less troublesome when analyzing single sperm cells, since these are haploid, showing only one allele peak per locus on an electropherogram.

The successful STR analysis of single micromanipulated sperm cells after application of the Repli-g single-cell MDA kit has been demonstrated in a recent study in which individual sperm cells were isolated using an adhesive-coated tungsten needle tip [84]. Consensus profiles were obtained by analyzing two different dilutions of the STR multiplex PCR products following WGA, and by this means the majority of single sperm cells yielded more than 80% of alleles of the haploid profiles, and several single sperm cells yielded full haploid profiles. Furthermore, gonosomal STR profiles of the single sperm cells were successfully analyzed as well and helped to compile the diploid autosomal STR donor profile from single sperm cells. The study also successfully analyzed single sperm cells from mock vaginal swabs with one or two male contributors, and thus the application of Repli-g MDA was suggested for sexual assault cases (or archival material) with low sperm counts or for multiple rape cases. An advantage over other approaches of single-sperm-cell STR analysis, such as low volume on-chip PCR [81] would be that, apart from the micromanipulation, all steps can be carried out with standard equipment of forensic laboratories. The WGA enrichment of template DNA would furthermore allow for replicate analysis or subsequent analysis of additional markers, if required.

## 6. Conclusions

Despite the shortcomings of classical WGA methods in forensic STR analysis, the latest single-cell WGA methods have opened up a new perspective for WGA in mixture deconvolution based on the analysis of single cells or small bioparticles. Methodologically, the successful application of WGA in the analysis of single sperm cells has already been demonstrated and now awaits further validation using forensic real-world samples. Applications of single-cell WGA to other bioparticles or to single cells (or 2-cell subsamples) micromanipulated from forensic traces still need to be tested. Furthermore, stimulated by biomedical interests (such as liquid biopsy and preimplantation genetic testing [85,86]), several commercial single-cell WGA kits have entered the market, and novel, low-bias single-cell methods have been developed which may turn out useful in a forensic context. Finally, forensic DNA analysis is in the process of implementing high-throughput sequencing methods, allowing for expansion of the sets of markers analyzed and for analysis of shorter stretches of DNA throughout [87]. In that respect, it should be noted that WGA in itself is not yet a DNA typing analysis, and the actual genotyping of forensic markers is carried out thereafter. Thus, WGA will leave the legal admissibility or the biostatistical properties of a chosen marker set untouched. By ideally uniformly amplifying the entirety of genomic DNA, WGA methods are open for any particular DNA marker type; however, different WGA methods may be better suited for particular types of markers [88,89]. It will be interesting to see in how far WGA methods might be compatible with or improve upcoming methods of forensic DNA analysis.

## Figures and Tables

**Figure 1 ijms-23-07090-f001:**
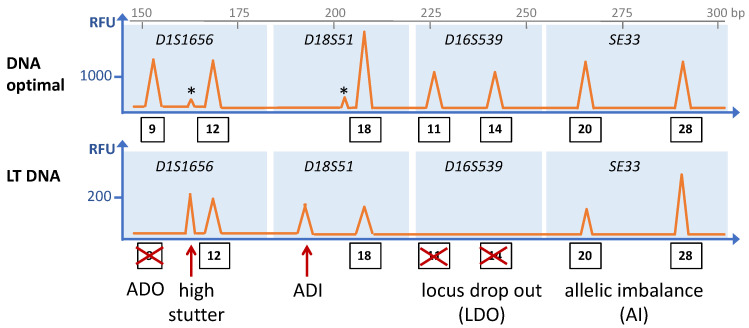
Schematic depiction of an electropherogram of an STR analysis showing the common stochastic effects of low template (LT) DNA (**bottom** panel) as compared to optimal DNA amounts (**upper** panel) from the same individual. The STR loci are indicated in italics, and the blue rectangles encompass the size ranges of amplicons of the particular STR loci. Thus, peaks within that range are assigned to the respective STR loci, and numbers in rectangles indicate the allele numbers of the peaks. Asterisks indicate stutter peaks. ADO: allele drop-out; ADI. Allele drop-in. The fragment length in base pairs (bp) is indicated on the x-axis on top, and the y-axis represents the peak height in relative fluorescence units (RFU). Please note the different scale in RFUs between the two panels.

**Figure 2 ijms-23-07090-f002:**
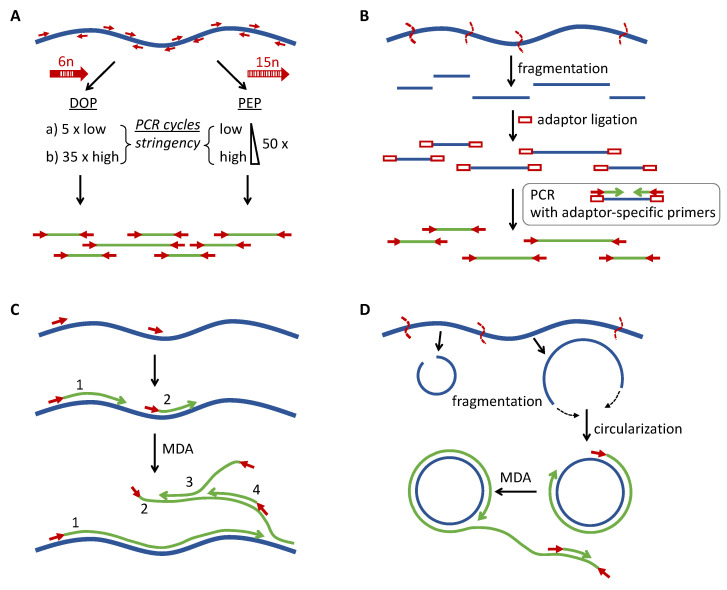
The most commonly applied WGA principles. (**A**) PCR using primers (red arrows) with randomized sequences. DOP-PCR uses primers in which 6 random nucleotides (6n; open rectangles) are flanked by fixed sequences (filled parts of the arrow), and the PCR program consists of a few low-stringency cycles, followed by high stringency cycles that specifically amplify the products of the low stringency step. PEP PCR uses completely randomized primers (15-mers, 15n), and the annealing temperature is continuously raised in each PCR cycle. (**B**) Adaptor ligation-PCR. The template DNA is fragmented, and after ligation of adaptor oligonucleotides to both sides, the fragments are amplified using adaptor-specific primers (red arrows). (**C**) Isothermal amplification by MDA. After denaturation, random hexamer primers anneal to the template DNA and are elongated using phi29 polymerase. Preceding double strands will be displaced by phi29, generating novel single-stranded templates for further primer annealing and extension. By this means, DNA can be amplified at a constant temperature of 30 °C. The numbers refer to the order of strand displacements. (**D**) Rolling circle variation of MDA. Fragmented DNA is circularized by fill-in and ligation. Following denaturation, random primers hybridize to the circularized template DNA, leading to strand displacement after one round of phi29-mediated DNA synthesis, generating long chains of single stranded DNA containing multiple copies of the circular DNA that may serve as templates for further MDA reactions.

**Figure 3 ijms-23-07090-f003:**
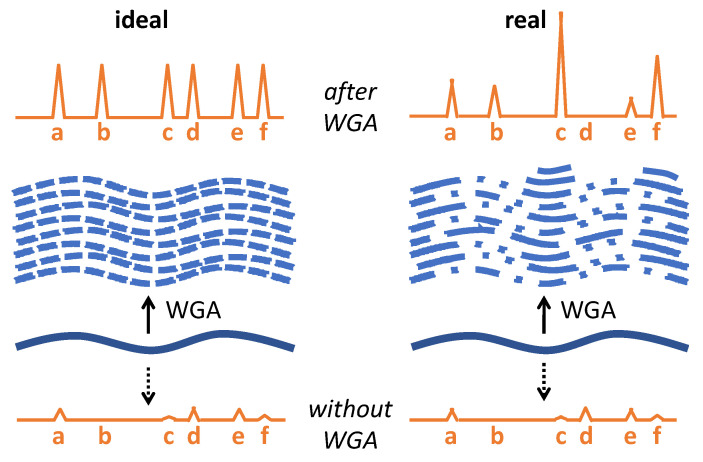
Consequences of WGA-inherent bias for STR typing. On the left side, the ideal situation is schematically depicted: WGA leads to even amplification of the complete template DNA, and the electropherogram shows a complete and balanced profile with high allele peaks (**a**–**f**), as compared to the incomplete LT DNA profile obtained without WGA (bottom). On the right, the real situation is depicted with uneven amplification, and consequently an imbalanced profile with ADOs (peak d). Please note that ADOs and AIs in LT DNA analysis without prior WGA are mainly due to stochastic sampling effects, whereas WGA-inherent bias causes additional AIs and ADOs.

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
