# Peer review of "New Perspectives for Whole Genome Amplification in Forensic STR Analysis"

_ijms, 2022, doi:10.3390/ijms23137090_

Round 1

Reviewer 1 Report

Several years ago we used WGA methods on small population samples and found that SNP typing was possible and seemingly accurate on the amplified material.  The review in this paper seems to cover the various WGA methods I am aware of.  I have one caveat that the authors might note when it comes to forensic results that may make it into court.  Any additional intervention between the raw sample and the results is a potential weakness that will need considerable justification or the defense will raise significant doubt.  I think there need to be major studies of the accuracy for forensic use.  That is not what this review is about, but the point might be emphasized if the authors want.

There are several individually minor points

Line 52-3 there are up to six fluors sometimes used.

Line 64 number agreement: type – artifacts

Line 68-9 a plus 1 stutter can also occur

Figure 1 and caption do not explain the term, LT DNA; similarly ADO and ADI are not defined. All are defined in the text around lines 98 to 102, but this comes after reference to the figure.  The definitions should come before the figure.

Line 112 words missing or garbled: “thus information be better”

Line 117 an overstatement of the expected result somewhat walked back in the following paragraph

Line 170 “of a few low”

Line 209 “the way that the random”

Line 256 “later on.”

Line 268-73 This whole paragraph is one sentence that is not grammatical.  It should be broken up into about 3 sentences.

Line 283-5 The sentence is not grammatical as written.  The “however” is not used correctly (preceded by a period or semicolon and/or followed by a comma) and the “clause” following has no subject for the verb “noticed”. Simply putting a comma after the “however” eliminates the confusion.

Reviewer 2 Report

The topic is reviewed in-depth and applicable to the practice of forensic genetics. I hope the figures used in this review are original figures.

Abstract-Line 2-"tiniest traces": Indicate to the reader the quantity that you are referring to while mentioning "tiniest traces".

Abstract-2nd sentence: This needs to be rephrased. Similarly, a few sentences across the manuscript need to be rephrased for clarity.

Provide a heading titled "Introduction" before the heading on "Forensic DNA profiles". Introduce to the reader the flow of the different sub-topics covered in this review and conclude the "introduction" section with the objective of this review. 

Reviewer 3 Report

This is a very complete work, so I would like to congratulate the authors in this regard.

The main problem I see is the total absence of mention of the legal issues that exist in forensics to amplify the complete genome. In most countries there is legislation that completely restricts the complete analysis of the human genome, being only possible to analyze the non-coding region. This is never mentioned by the authors.

On the other hand, the authors lament the fact that forensic geneticists have "abandoned" analysis or lost interest in whole genome amplification. However, for forensic questions, is it really necessary to know the entire genome to carry out an identification? What advantages does it bring over the current polymorphic markers? Would the value of LR increase in the determination of kinship? There is a missing section where a balance is made between the advantages and disadvantages in relation to current forensic markers.

Round 2

Reviewer 3 Report

I consider that the article is now ready for publication.